# Does Shoot Age Influence Biological and Chemical Properties in Black Currant (*Ribes nigrum* L.) Cultivars?

**DOI:** 10.3390/plants11070866

**Published:** 2022-03-24

**Authors:** Boban Djordjević, Dejan Djurović, Gordan Zec, Dragana Dabić Zagorac, Maja Natić, Mekjell Meland, Milica Fotirić Akšić

**Affiliations:** 1Faculty of Agriculture, University of Belgrade, Nemanjina 6, 11080 Belgrade, Serbia; dejan.djurovic@agrif.bg.ac.rs (D.D.); zec@agrif.bg.ac.rs (G.Z.); fotiric@agrif.bg.ac.rs (M.F.A.); 2Innovation Centre, Faculty of Chemistry Ltd., Studentski trg 12–16, 11000 Belgrade, Serbia; ddabic@chem.bg.ac.rs; 3Faculty of Chemistry, University of Belgrade, 11158 Belgrade, Serbia; mmandic@chem.bg.ac.rs; 4NIBIO Ullensvang, Norwegian Institute of Bioeconomy Research, Ullensvangvegen 1005, N-5781 Lofthus, Norway; mekjell.meland@nibio.no

**Keywords:** anthocyanin, flavanols, fruit quality, phenology, phenols, principal component analysis, yield

## Abstract

The aim of this study was to examine the influence of shoot age on the biological and chemical properties of 13 black currant cultivars with different origins and ripening times. Phenological observations together with examined pomological and chemical characteristics were studied in two consecutive years at the experimental field near Belgrade, Serbia. The total content of phenols was estimated spectrophotometrically by the Folin-Ciocalteu method, while quantitative analysis of anthocyanin and flavonols aglycones was performed using a high-performance liquid chromatographic (HPLC) method. Principal component analysis was performed to establish differences in biological and chemical properties of black currants. Three-year-old shoots had an earlier start of all examined phenological stages, better generative potential, higher yields, while clusters and berries from 2-year-old shoots had significantly higher values for physical properties, total phenols, anthocyanin and flavanols aglycones and antiradical capacity. Late ripening cultivars had higher contents of all chemical compounds. The berries on 2-year-old shoots had total phenolics that ranged between 123.0 (‘Titania’) and 298.3 mg/100 g fresh weight (FW) (‘Ometa’), while total anthocyanins ranged between 398.5 (’Ojebyn’) and 1160.8 mg/kg FW (’Ometa’). According to the obtained results, cultivars ‘Ometa’, ‘Ben Lomond’, ‘Tsema’ and ‘Malling Juel’ can be recommended as the most promising for growing in the continental climate because they stood out with higher generative potential and yield, physical traits of cluster and berry, higher level of primary and secondary metabolites and DPPH activity in their berries.

## 1. Introduction

Black currant (*Ribes nigrum* L.) belongs to the *Ribes* genus which consists of nearly 150 diploid species of spiny and non-spiny shrubs. It is a perennial small bush, indigenous to Central and Northern Europe, Caucasus, Central Siberia and Himalaya. It is widely cultivated in Europe, North America, New Zealand, and China, as a garden shrub and as an important commercial crop [1,2]. The centres of genetic diversity for blackcurrant are in northern Scandinavia and Russia [3]. Domestication started 400 years ago in the UK, from where it was spread throughout Europe and later to the USA, China, and New Zealand. Yearly production is ~490,000 tones, and the majority of the production originates from Europe, where Poland is the top producer, followed by Ukraine and the UK [4]. Black currant cultivation is in a continuous expansion, with a huge importance in human nutrition and a high degree of suitability for industrial processing. Berries of black currants are traditionally used for producing juices, jams, purées, jellies, syrups and yoghurt and other dairy products, but can be also eaten raw [5]. Besides berries, the seeds and leaves are a rich source of numerous bioactive ingredients, especially polyphenolic compounds [6,7].

The size of bushes, ripening time, berry size, yield, and other quantitative and qualitative properties of the fruits differ between the various cultivars, as well as by the plant’s age, growing location, environment conditions, cultivation systems, different pre- and post-harvest factors, ripening stage of the berries, and the number of clusters [8,9,10]. An increased number of long annual shoots affects the yield and fruit quality of red currants, together with the percentage of the most suitable fruits for fresh marketing [11]. The development of shoots is influenced by environmental factors, such as soil temperature and moisture content, as well as by genetic factors such as heredity, and interactions between different parts of the plant. Root development depends on shoot carbohydrates, growth regulators and other organic substances, while shoot development depend on the roots’ absorption of water, minerals and growth regulators [12]. Large yield in currants is correlated with the vigour and the age of the shoots, flower bud formation in the previous season, and berry size [13,14]. Moreover, older shoots of bushes have better generative potential, higher yields, as well as higher total phenol and vitamin C contents in the fruits, compared with younger shoots [15].

Black currant fruits are known for their high concentration of vitamin C, which is essential for building up the immune system in humans. Black currants have higher levels of ascorbic acid compared to other fruit species, and the levels are relatively stable due to the presence of large amounts of phenolic compounds and anthocyanins, which act as protective agents of ascorbic acid [16]. The quality of the berries is significantly influenced by the content of the primary metabolites (mostly sugars and acids) which are responsible for the sweetness of the fruit [17]. The fructose, glucose and sucrose are the main sugars in berries, while citric and malic acids are the dominant acids [18]. Regarding the mineral profile, black currants are an exceptional source of K, Ca, Mg, Fe, and Cu [19]. 

Due to their high contents of secondary metabolites, such as polyphenols, tannins and vitamins, berries of black currants represent a great source of natural antioxidants. [20,21]. Phenolic compounds have been reported to contribute positively to human health due to their antioxidant, antimicrobial and anti-inflammatory effects [22]. In this way black currant extracts support the digestive, nervous, and circulatory systems, and inhibit the multiplication and growth of cancer cells. Black currants are recognized as a good source of polyphenols especially anthocyanins, flavonols, flavan-3-ols, and phenolic acid derivatives [23,24]. *Ribes nigrum* cultivars contain up to fifteen anthocyanin structures, including four compounds: delphinidin 3-O-glucoside, delphinidin 3-O-rutinoside, cyanidin 3-O-glucoside, and cyanidin 3-O-rutinoside, which account for more than 90% of the total anthocyanin content [25]. Cultivars of black currant contain glycosylated forms of flavonols, mainly myricetin, quercetin, and kaempferol. These compounds are involved in mechanisms of tolerance to many types of abiotic stress, including UV radiation, extreme temperatures, ozone exposure, drought, or salinity in soils [26,27]. The profiles of bioactive metabolites and flavour compounds are not constant in black currant berries [24]. The phenolic profiles and contents in berry fruits vary based on the genetic background, climate, growing conditions, agronomic practices, and post-harvest handling techniques [28,29,30].

From a review of the literature, and to the best of our knowledge, other than comparable research in red currants [15], no similar study in black currant cultivars has been performed to date. The goal of this study was to determinate if the cultivar and shoots age influence the biological and chemical properties of black currants grown in Serbia. Finally, the best cultivars for growing will be pointed out and recommend for the production in temperate or related climatic conditions.

## 2. Results

### 2.1. Environmental Measurements

The environmental parameters in 2018 and 2019 were different during the research period (Figure 1). In 2018 higher average values of humidity and precipitation, and lower values of temperatures compared to 2019 were recorded. During berry development (from May till the first weeks of June) in 2018 there was 28.5 mm more precipitation than in same period in 2019. Moreover, after harvesting and during the young shoot development stage in 2018, there was 40.8 mm more precipitation than in same period in 2019.

In both experimental years the highest average values of temperature were recorded in July and August. During berry ripening (during June and July) in 2019 average daily temperatures were 4.5 °C higher than same period in 2018. After harvesting and during young shoot development, average daily temperatures in 2019 were 5.2 °C higher than during the same period in 2018.

### 2.2. Phenological Observations

Compared with 2-year-old shoots, all examined phenological properties in 3-year-old shoots of all black currant cultivars began earlier (Table 1). In 2018, the cultivar ‘Malling Juel’ had the earliest bud burst (20 February), while the cultivar ‘Titania’ had the latest (1 March). The shortest time difference in bud burst between shoots was recorded in the cultivar ‘Bona’ (2 days), while the longest was in cultivars ‘Titania’ and ‘Malling Juel’ (7 days). In the first year (2018), the cultivar ‘Čačanska Crna’ began blooming the earliest, while the latest cultivar was ‘Ben Sarek’, which also had the longest time difference between 2- and 3-year-old shoots (7 days). The cultivar ‘Čačanska Crna’ was the earliest to reach full bloom, while the cultivar ‘Ojebyn’ was the latest. The time difference in this phenological phase between 2 and 3-year-old shoots was between 1 (‘Bona’, ‘Titania’ and ‘Ojebyn’) and 4 days (‘Tenah’ and ‘Tsema’). The cultivar ‘Bona’ was first to be ready for harvest (9 June), while the cultivar ‘Ometa’ was the last (5 July). The shortest time difference in the start of harvest between 2- and 3-year-old shoots was recorded in cultivar ‘Ben Sarek’ (3 days), while the longest was recorded in ‘Ometa’ (7 days). 

In 2019, all examined cultivars had a later onset of bud burst by 7 to 15 days (Table 1). The cultivar ‘Bona’ had the earliest bud burst (1 March), while the cultivar ‘Titania’ had the latest (16 March). The time difference for this phenological phase between 2- and 3-year-old shoots was shorter compared to 2018 years, between 1 (‘Ben Nevis’) and 5 days (‘Titania’). All cultivars started blooming in first weeks of April. Compared to 2018, in 2019 all cultivars showed a longer time span between bud burst and the beginning of blooming (the average for all cultivars in 2018 was 18 days, while in 2019 it was 26 days) and a shorter time span between beginning and reaching full bloom (the average for all cultivars in 2018 was 21 days, while in 2019 it was 9 days). The cultivar ‘Bona’ had the earliest times of beginning and full blooming (1 and 8 April, respectively), while the cultivar ‘Titania’ was the latest (12 and 17 April, respectively). In both 2018 and 2019, the cultivar ‘Bona’ had the earliest beginning of harvesting, and the cultivar ‘Ometa’ had the latest. Compared to the 2018 year, despite a later time of bud burst and full blooming in 2019, all cultivars were ready to be harvested earlier (by an average of four days). Moreover, the time difference between 2- and 3-year-old shoots for this phenological phase was shorter than in 2018, at between 2 (‘Silmu’) and 6 days (‘Tsema’).

### 2.3. Generative Properties

In both experimental years all cultivars had higher values of all physical properties of clusters and berries on 2-year-old shoots compared to 3-year-old shoots (Table 2a,b). In 2018, the cultivar ‘Bona’ had the highest cluster weight on 2-year-old shoots (13.4 g), while ‘Titania’ had the highest weight on 3-year-old shoots (9.9 g). Three cultivars (‘Ben Sarek’, ‘Bona’ and ‘Tsema’ had a statistically significant difference between the cluster weights of 2- and 3-year-old shoots. The cultivar ‘Tsema’ had the highest values for flower and berry number on both types of shoots, while ‘Ben Nevis’ had the lowest. The percentage of fruit set was between 59.1% (’Silmu’ on 3-year-old shoots) and 94.9% (’Titania’ on 3-year-old shoots). Interestingly, most of cultivars had higher rates of fruit set on 2-year-old shoots but no statistically significant difference between 2 and 3-year-old shoots was recorded. The cultivar ‘Bona’ had the highest berry weight (2.1 g and 1.5 g on 2- and 3-year-old shoots, respectively), while ‘Ometa’ had the lowest (0.9 g and 0.7 g on 2- and 3-year-old shoots, respectively). All cultivars had higher berry weight on 2-year-old shoots compared to 3-year-old shoots, but a significant difference was recorded in only four cultivars. All cultivars, except ‘Triton’, had marketable fruit diameter of berries (>12 mm) on 2-year-old shoots, while only three cultivars (‘Bona’, ‘Ben Nevis’, and Ben Sarek’) did on 3-year-old shoots. All cultivars showed significantly higher values of cluster number and total yield on 3-year-old shoots compared to 2-year-old shoots. The cultivar ‘Ben Nevis’ had the highest cluster number on 3-year-old shoots (67.1) and ‘Ojebyn’ had the lowest (28.3), while on 2-year-old shoots, the cultivar ‘Malling Juel’ had the highest cluster number (28.4) and ‘Ojebyn’ had the lowest (9.6). ‘Malling Juel’ had the highest total yield (2.55 kg/plant) while ‘Ojebyn’ had the lowest (0.75 kg/plant). The yield on 2-year-old shoots was between 0.08 kg (‘Ojebyn’) and 0.31 kg (‘Tsema’). The share of yield on 2-year-old shoots in total yield was between 20.8% (‘Ben Nevis’) and 40.0% (‘Tsema’). Among all cultivars, ‘Titania’ had the highest yield on 3-year-old shoots (0.57 kg), while ‘Ojebyn’ had the lowest (0.18 kg). 

In 2019, generally, all cultivars had lower values of examined physical traits. The only exceptions were length and number of flowers per cluster (Table 2b). Cultivar ‘Bona’ still had the highest cluster weight on 2-year-old shoots (11.3 g), while on 3-year-old shoots, ‘Tsema’ had the highest cluster weight (8.8 g). Moreover, eight cultivars had a significant difference in cluster weight between 2 and 3-year-old shoots. Cultivar ‘Tsema’ had the highest number of flowers and berries per cluster on both types of shoots, and was the only cultivar that showed a significant difference in flower number per cluster between 2 and 3-year-old shoots. In 2019 all cultivars had a higher number of berries per cluster on 2-year-old shoots, while five cultivars had a significant difference between shoots. In the second year decreased values of fruit set were recorded, especial on 2-year-old shoots, compared to the first year. In this regard it was also noticed that nine cultivars had higher values of fruit set on 3-year-old shoots compared to 2-year-old shoots, while only six had a significant difference. In 2019 in most of the examined cultivars’ berry weights and berry diameters were lower compared to 2018. Nevertheless, comparing 2- and 3-year-old shoots, only cultivar ‘Bona’ had a statistically significant difference in berry weight, while cultivars ’Ben Sarek’, ‘Bona’, ‘Ometa’, and ‘Triton’ had statistically different berry diameters. In the second year (2019), a decrease in cluster number per shoot, especially on 3-year-old shoots, influenced a decrease in total yield in eleven out of thirteen cultivars. ‘Tsema’ had the highest yield (2.58 kg), while ‘Ojebyn’ had the lowest (0.56 kg). On the other hand, an increase in cluster number per shoot, counting only 2-year-old shoots, influenced an increase in total yield per plant. The share of yield on 2-year-old shoots in total yield was between 21.2% (‘Ometa) and 45.2% (‘Tsema’).

### 2.4. Determination of Total Soluble Solids (TSS), Total Acids (TA), Total Sugars (TS) and Ascorbic Acid (AA)

The TSS content of berries varied significantly among the cultivars and shoot ages (Table 3). ‘Ometa’ had the highest content of TSS (16.1% on 2-year-old shoots and 13.6% on 3-year-old shoots), while ‘Bona’ had the lowest (10.5% and 8.9%, respectively). Also, all cultivars showed higher content of TSS in berries on 2-year-old shoots compared to 3-year-old shoots and significant differences were recorded in four cultivars (‘Silmu’, ‘Titania’, ‘Malling Juel’, and ‘Ojebyn’). The values of TA ranged between 1.1% (‘Čačanska Crna’) and 2.1% (‘Ben Sarek’). All cultivars except ‘Malling Juel’ and ‘Čačanska Crna’ had higher levels of TA in 2-year-old shoots. ‘Ometa’ had the highest content of TS 13.2% (2-year-old shoots) and 10.2% (3-year-old shoots). Moreover, all cultivars except ‘Ojebyn’ had higher content of TS in 2-year-old shoots compared to 3-year-old shoots, while the difference was statistically significant only between the shoots of ‘Ometa’ and ‘Čačanska Crna’. The sweetness (sugar–acid ratio) of berries was between 3.8 (‘Ben Nevis’) and 9.7 (‘Čačanska Crna’) and most of the cultivars had higher values on 2-year-old shoots. Cultivar ‘Čačanska Crna’ had the highest content of ascorbic acid (175.3 mg% on 2-year-old shoots, and 158.3 3 mg% on 3-year-old shoots). All cultivars, except ‘Ben Sarek’ and ‘Tenah’ had higher content of AA in berries on 2-year-old shoots, while statistical differences between shoot types were recorded in six cultivars. 

When compared to 2018, in 2019 an increase in TSS content in berries of 11 cultivars was recorded. Moreover, eight cultivars had significantly higher TSS content on 2-year-old shoots. In the second year, all cultivars had an increased content of TA in berries, which was higher in berries on older shoots. ‘Ometa’ had the highest values of TA and ‘Titania’ had the lowest. In 2019, most of the cultivars had an increase in TS, while the difference between berries on 2- and 3-year-old shoots was significant in seven cultivars. In all cultivars a higher content of TS on younger shoots was recorded. The berry sweetness in all cultivars decreased, especially on 3-year-old shoots. ‘Titania’ had the highest values of berry sweetness (6.9 on 2-year-old shoots and 4.8 on 3-year old shoots), while ‘Bona’ had the lowest (3.7 and 2.7, respectively). In 2019 a decrease in AA content compared to 2018 was recorded. ‘Malling Juel’ had the highest content of AA, while ‘Ojebyn’ had the lowest. However, individually, four cultivars had an increase in vitamin C content (‘Ben Lomond’, ‘Ometa’, ‘Titania’, and ‘Malling Juel’). In the second year most of the cultivars had higher content of ascorbic acid in berries from 2-year-old shoots, while ten cultivars had a significant difference between shoot ages.

### 2.5. Determination of Total Phenolics (TPC), Radical Scavenging Activity (DPPH), Total Anthocyanins Aglycones (TAA), and Total Flavonols Aglycones (TFA) Contents 

TPC of berries significantly varied among the examined cultivars and with shoot age (Table 4a,b). In both examination years all cultivars had higher content of total phenolics in berries on 2-year-old shoots. During 2018 the cultivar ‘Ometa’ had the highest TPC (202.3 mg/100 g FW on 2-year-old shoots and 184.2 mg/100 g FW on 3-year-old shoots), while ‘Titania’ had the lowest (123.0 mg/100 g FW and 111.2 mg/100 g FW, respectively). A significant difference in TPC between berries originated from 2- and 3-year-old shoots was determined in five cultivars. All cultivars, except ‘Triton’, had higher content of TAA on 2-year-old shoots. TAA in berries on 2-year-old shoots ranged between 765.8 mg/kg FW (‘Ometa’) and 398.5 mg/kg FW (‘Ojebyn’), while in berries from 3-year-old shoots had between 491.5 mg/kg FW (‘Ben Nevis’) and 293.9 mg/kg FW (‘Tenah’). Seven cultivars had a significant difference in TAA levels between berries from 2- and 3-year-old shoots. Among the samples investigated, berries of ‘Ben Lomond’ showed the highest DPPH radical scavenging activity with IC_50_ values of 2.6 mg/mL and 4.0 mg/mL, on 2-and 3-year-old shoots, respectively, while berries of ‘Titania’ had the least activity with IC_50_ values of 6.1. mg/mL and 7.2 mg/mL, on 2-and 3-year-old shoots, respectively. Moreover, among all cultivars studied, higher values of DPPH radical scavenging activity in berries from 2-year-old shoots compared to 3-year-old shoots were recorded, and significant differences between the two shoot types in cultivars ‘Ben Lomond’ and ‘Tsema’ were determined.

The most represented AnA in all cultivars was delphinidin 3-rutinoside (from 114.9 mg/kg FW in ’Tsema’ on 3-year-old shoots, to 289.2 mg/kg FW in ‘Ometa’ on 2-year-old shoots) followed by cyanidin 3-rutinoside (from 114.0 mg/kg FW in ‘Tenah’ on 3-year-old shoots to 301.2 mg/kg FW in ‘Ometa’ on 2-year-old shoots). Only ‘Triton’ had higher content of AnA in berries from 3-year-old shoots. Most of the cultivars showed a significant difference in content of AnA on 2-year-old shoots compared to 3-year-old shoots. In 2018 the highest content of TFA in berries on both types of shoots was in ‘Tsema’ (14.1 mg/kg FW on 2-year-old shoots and 12.3 mg/kg FW on 3-vear-old shoots), while the lowest level was in ‘Titania’ (7.3 mg/kg FW and 6.8 mg/kg FW, respectively). The most represented flavonoid aglycone in all cultivars was quercetin. No significant difference in levels of TFA in berries was recorded between the two types of shoots. 

In the second year, in almost all cultivars and in both shoot types, increased levels of phenolic compounds and DPPH activity were recorded. The highest values of TPC were in the cultivar ‘Ometa’ (298.3 mg/100 g FW in 2-year-old shoots and 275.3 mg/100 g FW in 3-year-old shoots), followed by ‘Malling Juel’ (254.3 mg/100 g FW and 222.2 mg/100 g FW). Higher values of TPC in berries from 2-year-old shoots compared to 3-year-old shoots were recorded in all cultivars, but the differences were not statistically significant. The highest values of TAA were obtained from berries of the ‘Ometa’ cultivar (1160.8 mg/kg FW in 2-year-old shoots and 803.3 mg/kg FW in 3-year-old shoots), followed by ‘Ben Sarek’ (753.6 mg/kg FW and 687.3 mg/kg FW respectively). In all cultivars studied, TAA values were higher in berries from 2-year-old shoots but a significant difference between shoot ages was determined only for cultivars ‘Ometa’ and ‘Ben Nevis’. In 2019, berries from the cultivar ‘Ben Lomond’ showed the strongest DPPH radical scavenging activity with IC_50_ values of 2.1 mg/mL (2-year-old shoots) and 2.3 mg/mL (on 3-year-old shoots), while the lowest activity was recorded for berries of the ‘Ojebyn’ cultivar with IC_50_ values of 4.1 mg/mL and 4.3 mg/mL, respectively. Overall, higher values of DPPH radical scavenging activity were recorded in berries from 2-year-old shoots, although only cultivars ‘Malling Juel’ and ‘Tsema’ had statistically significant differences between shoot types.

In 2019, all cultivars had higher contents of delphinidin 3-rutinoside in berries from 2-year-old shoots than from 3-year-old shoots, although significant differences between shoot types was only observed in cultivar ‘Čačanska Crna’. However, in the second year the values of cyanidin 3-rutinoside were higher in berries from 3-year-old shoots. In the same year, cultivars ‘Bona’ and ‘Ben Nevis’ had the highest total flavonol contents in berries from both types of shoots, while ‘Titania’ and ‘Triton’ had the lowest. As in the first year, no significant differences in flavonol compounds between both shoot types were recorded.

### 2.6. Principal Component Analysis (PCA)

A data matrix of 13 (the number of black currant samples) × 31 (physical attributes of clusters and berries, quantified phenols, total phenols, total anthocyanins, total flavonoids, total soluble solids, total acids, total sugars, sweetness, and ascorbic acid) was processed using a covariance matrix with autoscaling. PCA resulted in a four-component model which explained 76.38% of the total variance (PC1, PC2, PC3, and PC4 accounted for 31.82%, 17.82%, 14.97%, and 11.78%, respectively). The PCA score plots for the first two principal components presented in Figure 2A show trends, groupings, and outliers. In Figure 2A it is evident that the investigated black currant samples were clustered in two groups. For all cultivars, three-year-old shoots were separated from the 2-year-old shoots along the PC1 axis. The PC score plot showed that sample OM3 (‘Ometa’—3-year-old shoots) could be considered as an outlier (exceeded the limits imposed by the Hotelling’s T2 95% probability ellipse). From the loading plot of PCA scores (Figure 2B), it was possible to identify the most influential variables responsible for the separation among samples.

## 3. Discussion

The beginning of the vegetation period and the durations of phenological phases of black currant cultivars are caused by the genotype and the environmental conditions of the agricultural habitat. Temperature is a major driver for phenological events, but little information is available about the effects of a winter chilling period on the plants’ development. In addition, more genes are putatively associated with dormancy release, growth or development of black currants cultivars. However, recent research has determined the abundance of 32 selected genes which are associated with overwintering or spring phenology in black currants [31]. In all cultivars and both years, earlier beginning of bud burst was recorded in 3-year-old shoots compared with 2-year-old shoots. This was probably due to the fact that the old shoots possessed higher levels of nitrogen and carbohydrate storage, which caused an earlier start of vegetation, as already proved in [32]. The ‘Čačanska Crna’ cultivar had the earliest onset of flowering and full blooming, which is consistent with previous research on black currants [33]. The cultivars which had the higher number of clusters per plant had the largest difference in harvesting times when shoot ages were compared. This is probably due to the genotype, especially the vigour of the bush [15]. Despite a later start to the vegetative period, all cultivars had an earlier ripening time in the second year, probably as a result of higher temperatures during the development stages of the berries [29].

The yield in black currants is mainly determined by the environmental conditions during floral initiation in the previous season; therefore, berry yield is not affected by post-flowering environmental conditions [34]. The studied cultivars showed significantly higher results in terms of cluster length, berry weight, number of flowers, and berries per cluster on 2-year-old shoots, a finding which confirms previous reports in a study on red currants [15]. The cluster number per bush and cluster weight are the most responsible traits important for yield [15,29,30,31]. Many morphological features of vegetative organs and berries depend on the physiological state of the plant, agro-technical practices, and weather conditions during the growing season [35]. The number of berries per cluster and the size of the berries were much higher than those observed by other authors who examined the same cultivars [18,36]. The differences between the results could be attributed to the different growing practises and the different environmental conditions. Additionally, the size potential of currant berries seems to be determined, among other things, by the position on the bunch, the conditions during flowering, fruit set, and the number of berries per bunch, together with biotic stresses and the nutrition and carbohydrate status of the plant [8,30]. 

Numerous factors such as cultivar, ecological conditions, maturity stage, and crop load can influence the fruit’s chemical composition [37]. Higher values of total soluble solids are mostly connected with low crop load [38]. Late-ripening cultivars show higher contents of TSS in berries compared to early-ripening cultivars in both types of shoots [39]. TSS concentration is positively correlated with temperature before harvest, and negatively with precipitation [40]. The results of this study are in agreement with the previously mentioned studies since higher values of TSS were observed in the second year, when warmer weather conditions were recorded. The sugars in black currants’ fruit are mainly mono- and disaccharides and the relative proportions of these individual and total sugars are important for the perception of sweetness [41]. Moreover, ascorbic acid is an important chemical quality component in black currant berries. In this study, significant variations in ascorbic acid concentration among cultivars and years were recorded. The observed decrease in ascorbic acid concentration in the second, warmer, season is in agreement with a previous study performed in a controlled climate [40].

The polyphenol content and antioxidant activity of currants could be affected by ecological factors, the degree of ripeness, and soil conditions [42]. Late-ripening black currant cultivars had higher contents of total phenols, while earlier ones had lower contents of phenolics due to their shorter maturity period [43]. In this experiment, phenolic contents of berries varied between harvest dates. Cultivars with higher shoot lengths had higher leaf weights (data not shown). The assimilation area strongly affected berry quality, colour, aromatic compounds, and secondary metabolites. Previous reports have pointed out the importance of reaching and retaining the optimal ratio between the assimilation area and berry yield for an increasing phenolics content [37,38,44,45]. The differences in phenolic components during different growing seasons are most likely due to the different air temperature and rainfall rate, given the well-known inverse correlation between phenolic concentrations and air temperature. Plants which grown in warm climates very often have higher antioxidant properties, because production of these compounds is a strategy to counter the oxidative stress [46]. Content of total phenols ranged from 117.2 mg/100 g F.W. to 298.3 mg/100 g F.W. and these values were higher compared to results elsewhere in the literature [18,47]. In addition, seven of the examined cultivars had total phenol values over 200 mg/100 g of F.W, which is much higher than previously published results and berries of these cultivars have even higher total phenol contents than the fruits of certain cultivars of chokeberry [48,49,50]. However, the amount of total phenolics was smaller in our samples than in cultivars analysed by other authors [51]. 

The anthocyanin synthesis depends on genetic and environmental factors. Fruits from the older plants contained less anthocyanins, and a significant negative correlation between berry size and anthocyanin content of red currant cultivars has been previously determined [40,52]. Delphinidin 3-rutinoside and cyanidin 3-rutinoside were the most ubiquitous anthocyanin aglycones, and their contents varied among cultivars and between shoot ages. The contents of these anthocyanin aglycones ranged between 114.9 mg/kg and 289.2 mg/kg and were similar or slightly higher compared to previously obtained data [18,53]. The most represented flavanol aglycone in all cultivars was quercetin, and levels were two- to threefold higher than those observed by other authors [42]. Anthocyanin biosynthesis is regulated in vivo by a number of structural and regulatory genes. The structural genes directly involved in the biosynthesis of anthocyanins are chalcone synthase, chalcone isomerase, flavanone 3-hydroxylase, anthocyanidin synthase, and flavonoid 3-O-glycosyltransferase [54]. The regulatory genes encode members of several transcription factor families and include R2R3 MYB transcription factors [55]. Recent studies have shown that two different MYB transcription factor genes induce anthocyanin accumulation in fruit skins [56,57]. Sunlight radiation is a major environmental factor which affects anthocyanin synthesis by regulating gene expression and plant development, while in [58], ultraviolet-B was a major factor in increasing the synthesis of anthocyanins and flavonoids. In addition, stability of the anthocyanin aglycones depends on the type of anthocyanin pigment, the co-pigments, weather conditions, enzymes, and oxygen [59].

### Principal Component Analysis (PCA)

Principal component analysis was performed to establish differences in biological and chemical properties of black currants based on the cultivar and shoots’ age. Previously, PCA was employed successfully to separate currant cultivars according to many studied traits [60,61,62,63,64]. According to the plots (Figure 2A,B) it was obvious that bud burst (1), beginning of blooming (2), full blooming (3), length of shoot (12), number of clusters per shoot (13), yield per shoot (14), and % of shoot yield in total yield (15) were the most important factors for the separation of the 2-year-old shoots from the 3-year-old shoots (Figure 2B). On the other hand, the 3-year-old shoot samples were characterized notably by berry weight (10), berry diameter (11), kempferol (23) and quercetin (25) levels, and total flavonoids (26). Due to this, the negative part of the first principal component (PC1) can be considered the ‘phenology and shoot’ component, while the positive part of the PC1 can be considered the ‘berry and polyphenol’ component. This shows that phenology and cluster traits are positively correlated and can predict each other, while being negatively correlated with berry traits and polyphenol compounds.

Higher levels of total phenols (16), total anthocyanins (17), all quantified individual anthocyanins (18, 19, 20, and 21), total soluble solids (27), and total sugars (29) made sample OM3 (‘Ometa’, 3-year old) an outlier. The BO3 sample (‘Bona’—3-year-old shoots) was separated from other samples based on the highest values of cluster weight (5), berry weight (10), and berry diameter (11). The latest beginning of harvest (4) was the most important variable separating sample OM2 (‘Ometa’, 2-year-old shoots) from the other black currants. Generally, from the obtained results, ‘Ometa’ stood out as a cultivar rich in health-related compounds that can be grown to obtain functional food, while ‘Bona’, which is characterized by large berries, could be used for fresh consumption. 

## 4. Materials and Methods

### 4.1. Plant and Berries

In this study, 13 black currant cultivars: ‘Ben Sarek’ (BS), ‘Ben Nevis’ (BN), ‘Ben Lomond’ (BL), ‘Ometa’ (OM), ‘Tenah’ (TE), ‘Triton’ (TR), ‘Silmu’ (SI), ‘Tsema’ (TS), ‘Bona’ (BO), ‘Malling Juel’ (MJ), ‘Ojebyn’ (OJ), and ‘Čačanska Crna’ (CC) were evaluated over two consecutive years (2018–2019). The Experimental field was at the village of Mislođin, near Belgrade, Serbia, between 44°30′ and 44°45′ north latitude, at an altitude between 80 and 90 m. Mislođin is located near the centre of the northern warm temperate belt, with a milder climate that is more continental than the typical Pannonia. The average annual temperature in this area is about 11.3 °C, and annual rainfall is about 620 L of water/m^2^. Planting was carried out in spring 2007 with 1-year-old nursery plants at a distance of 1.8 m between rows and 0.8 m within the row, on sandy loam soil type, with an average pH of 6.1. The orchard was managed according to the integrated cultivation system. During the drought season, drip irrigation was used. During vegetation, contact herbicides were used for weed control. At the beginning and end of the growing seasons, mineral fertilizer (14% N, 7% P, 17% K) was applied in amounts of 200 kg/ha and 100 kg/ha, respectively. Cattle manure was applied every third year (10.0 kg/m^2^). The experiment was organised randomly with five repetitions (five bushes/repetition). On each bush, three two-year-old and three three-year-old shoots were left after winter pruning. Berries were hand-harvested in June–July, depending on the commercial ripening time for each cultivar (90% fruit surface colour). The environmental measurements were performed every day at 14 h. Measurements included the air temperature, relative humidity, and daily precipitation. The air temperature and relative humidity were measured at a height of 2 m using automatic weather stations MeteosCompact, Pesslinstruments GmbH, Austria, set in the immediate vicinity of the planting.

### 4.2. Phenological Observations

According UPOV Code for Ribes_Nig [65] the following phenological properties were studied: bud burst—when 10% of total buds showed bud burst, beginning of blooming—10% of flowers are open, full blooming—when 90% of flowers are open, and beginning of harvest—when the fruit can be easily removed from the plant). For all cultivars, the beginning of bud burst, beginning and full blooming, and beginning of harvest were recorded separately on two- and three-year-old shoots. 

### 4.3. Generative Properties

The following generative traits were evaluated: Cluster weight, cluster length, number of flowers per cluster, number of berries per cluster, berry weight, and fruit set (number of fruits per 100 flowers), number of clusters per shoot, and yield per shoot and bush. The weights of clusters and berries were expressed in g and cluster length was measured with a ruler and expressed in cm. Measurements included both types of shoots.

### 4.4. Determination of Soluble Solids, Titratable Acidity, Sugars, Sweetness, and Ascorbic Acid

After harvesting, the berries were stored at 3 °C and analysed within 24 h. In total, 500 g of undamaged berries were selected and manually crushed to obtain the juice. For analyses, juices were centrifuged at 14,000× *g* for 20 min. Total soluble solids (TSS) content of berries was determined using by refractometer (Atago, pocket PAL-1. Kyoto, Japan). Titratable acidity (TA) was determined by titrating the berry juice with 0.1 N NaOH up to pH 7.0. Acidity was expressed as percent of malic acid. Total sugars (TS) were determined by Luff–Schoorl method in % [66]. The sweetness of berries was determined as the ratio of total sugar content and total acids. Ascorbic acid was quantified using a reflectometer set from Merck Co. (Merck RQflex, Darmstadt, Germany). Fruit samples (5 g) and 20 mL oxalic acid (1%) were mixed, homogenized for 1 min and filtered. Polyvinyl-polypyrrolidone (PVPP) (500 g) was added to 10 mL of the filtered sample to remove phenols, and 6–7 drops of H_2_SO_4_ (25%) were added, to reduce the pH level below 1.14. Results were expressed as mg ascorbic acid 100 g^−1^ FW. Primary metabolites were analysed in the whole fruit without seeds. For each cultivar, three repetitions per sampling date were carried out. 

### 4.5. Determination of Total Phenolic Content 

The total content of phenols was estimated by Folin–Ciocalteu method with slight modifications [67]. Berries (10 g) were extracted with MeOH for 30 min in an ultrasonic bath and then filtered. Two hundred microliters of extracts were added to 1 mL of 1:10 diluted Folin–Ciocalteu reagent. After 4 min, 800 μL of sodium carbonate (75 g L^−1^) was added. After two hours of incubation at room temperature, the absorbance at 765 nm was measured. Gallic acid (0–100 mg L^−1^) was used for calibration of a standard curve. The results were expressed as milligrams of gallic acid equivalent per 100 g of fresh weight (mg GAE 100 g^−1^ FW).

### 4.6. Quantitative Analysis of Anthocyanin and Flavonols Aglycones 

Extraction of berries was performed as described in [8], with slight modification. Frozen berry samples were ground to a fine paste in a mortar chilled with liquid nitrogen and 5 g was extracted with 10 mL methanol containing 3% (*v*/*v*) formic acid and 1% (*w*/*v*) 2,6-di-tert-butyl-4-methylphenol (BHT) in a cooled ultrasonic bath for 1 h. BHT was added to the samples to prevent oxidation. After extraction, the berry extracts were centrifuged for 10 min at 10,000 rpm. Each supernatant was filtered through a Chromafil AO-20/25 polyamide filter produced by Macherey-Nagel and transferred to a vial prior to injection into the HPLC system. Phenolic compounds were analysed on a Thermo Finnigan Surveyor HPLC system (Thermo Scientific, San Jose, CA, USA) with a diode array detector at 350 nm (flavonols) and 530 nm (anthocyanins). Spectra of the compounds were recorded between 200 and 600 nm. The column was a Gemini C18 (150 mm × 4.6 mm 3 μm; Phenomenex, Torrance, CA, USA) operated at 25 °C. The elution solvents were aqueous 1% formic acid (A) and 100% acetonitrile (B). Samples were eluted according to the linear gradient: 0–5 min, 3–9% B; 5–15 min, 9–16% B; 15–45 min, 16–50% B; 45–50 min, 50% isocratic; and finally, washing and reconditioning of the column. The injection amount was 20 μL and flow rate was 1 mL min^−1^. All phenolic compounds were identified using a mass spectrometer (Thermo Scientific, LCQ Deca XP MAX) with electrospray ionization (ESI) operating in negative (all phenolic groups except for anthocyanins) and positive (for anthocyanins) ion modes. The analyses were carried out using full scan data-dependent MS^n^ scanning from *m*/*z* 115 to 1500. The injection volume was 10 μL and the flow rate was maintained at 1 mL min^−1^. The capillary temperature was 250 °C; the sheath gas and auxiliary gas were 20 and 8 units, respectively; and the source voltage was 4 kV for negative ionization and 0.1 kV for positive ionization. Spectral data were elaborated using the Excalibur software (Thermo Scientific). The identification of compounds was confirmed by comparing retention times and their spectra as well as by adding the standard solution to the sample and by fragmentation. Concentrations of phenolic compounds were calculated from peak areas of the sample and the corresponding standards and expressed in mg kg^−1^ fresh weight (FW) of berries. The total anthocyanins and flavonols contents were obtained as the sum of the individual aglycones 

### 4.7. DPPH Radical Scavenging Activity

The free radical scavenging activity of berries on the stable1, 1-diphenyl-2-picrylhydrazyl (DPPH) radical was carried out according to the procedure described previously in [68], with slight modifications. The juice samples (2 g) were diluted with distilled water up to 10 mL, centrifuged for 10 min, and the supernatant was used for analyses. The antiradical capacity of each extract was evaluated using a dilution series, in order to obtain a large spectrum of sample concentrations. The extracts (100 μL) were mixed with 1400 μL of 80 μM methanolic solution of DPPH. Absorbance at 517 nm was measured after 20 min. The percentage of inhibition was calculated using equation: Inhibition = [(A_0_ − A_i_)/A_0_] × 100
where A_0_ is absorbance of the control and A_i_ is absorbance of the samples. IC_50_ values were estimated using a nonlinear regression algorithm. Trolox was used as a positive control.

### 4.8. Statistical Analysis

The physical attributes of cluster were presented as the mean of five repetitions (twenty clusters per repetition) and for chemical traits, as triplicate determinations. Generative properties of cultivars presented as the mean of five repetitions (six bushes/repetition). One-way analysis of variance was performed using the STATISTICA 11.0 software package. The significant differences between means were determined with Fisher’s least significant difference post hoc test (LSD) at *p* < 0.05, level. 

Principal component analysis was carried out using the PLS_Tool Box software package for MATLAB (Version 7.12.0), Budapest, Hungary [69].

## 5. Conclusions

To the best of our knowledge, the current study is the first comprehensive analysis of biological and chemical properties in berries from shoots of different ages in black currant cultivars with different origins and ripening times. All examined cultivars had earlier beginnings of all phenological stages and significant higher yields in 3-year-old shoots, compared to the same properties in 2-year-old shoots. However, most of the cultivars had better physical attributes of clusters and berries, higher TSS, and higher content of phenolic compounds, ascorbic acid, and DPPH radical scavenging activity in younger shoots. Regarding the obtained results, cultivars ‘Ometa’, ‘Ben Lomond’, ‘Tsema’, and ‘Malling Juel’ stood out with later blooming, higher generative potential and yield, physical traits of clusters and berries, higher levels of primary and secondary metabolites, and DPPH activity in their berries. These cultivars are recommended as suitable for growing in Serbian or similar agro-ecological (temperate) conditions. The cultivar ‘Ometa’ was distinguished by its high content of secondary metabolites that can have a positive influence on human health and can be recommended for the production of functional food, while the ‘Bona’ cultivar, singled out due to having the largest berries, can be recommended for fresh consumption. All the mentioned cultivars could also be used for a subsequent breeding program in order to improve black currant cultivars. 

From a practical point of view, the obtained results make it possible for producers to plan agrotechnical practices and project fruit load, in order to obtain higher yields of berries that have better physical properties and higher levels of health-promoting phytochemicals.

## Figures and Tables

**Figure 1 plants-11-00866-f001:**
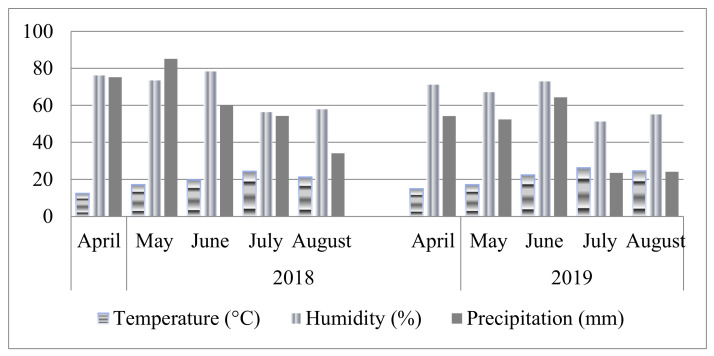
Average values of environmental conditions.

**Figure 2 plants-11-00866-f002:**
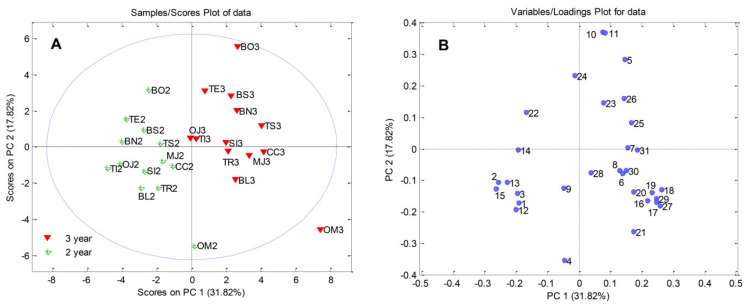
PC1–PC2 Score plots. (**A**) Scores and (**B**) loading plot of the traits studied in 2- and 3-year-old shoots of 13 black currant cultivars. Please see the Appendix A for the list of abbreviations.

**Table 1 plants-11-00866-t001:** Phenological phases of black currant cultivars during 2018 and 2019 year.

Cultivar	Age of Shoot	2018	Bud Burst	Beginning of Blooming	Full Blooming	Beginning of Harvesting	2019	Bud Burst	Beginning of Blooming	Full Blooming	Beginning of Harvesting
Ben Sarek	2		28.02.	25.03.	10.04.	21.06.		07.03.	03.04.	14.04.	19.06.
	3		24.02.	18.03.	08.04.	18.06.		05.03.	01.04.	11.04.	15.06.
Ben Nevis	2		01.03.	22.03.	12.04.	27.06.		07.03.	05.04.	15.04.	25.06.
	3		26.02.	17.03.	10.04.	23.06.		06.03.	02.04.	10.04.	20.06.
Bona	2		24.02.	16.03.	02.04.	14.06.		03.03.	03.04.	16.04.	14.06.
	3		22.02.	14.03.	01.04.	09.06.		01.03.	01.04.	08.04.	09.06.
Ben Lomond	2		03.03.	20.03.	07.04.	30.06.		10.03.	05.04.	16.04.	26.06.
	3		27.02.	14.03.	04.04.	26.06.		06.03.	01.04.	10.04.	23.06.
Ometa	2		02.03.	21.03.	14.04.	12.07.		08.03.	03.04.	14.04.	01.07.
	3		26.02.	16.03.	11.04.	05.07.		05.03.	01.04.	11.04.	27.06.
Tenah	2		01.03.	21.03.	14.04.	26.06.		08.03.	03.04.	14.04.	20.06.
	3		26.02.	18.03.	10.04.	24.06.		06.03.	02.04.	11.04.	16.06.
Silmu	2		01.03.	19.03.	12.04.	23.06.		09.03.	06.04.	15.04.	20.06.
	3		25.02.	16.03.	10.04.	18.06.		06.03.	04.04.	12.04.	18.06.
Titania	2		08.03.	18.03.	11.04.	24.06.		21.03.	07.04.	17.04.	20.06.
	3		01.03.	16.03.	10.04.	20.06.		16.03.	03.04.	12.04.	16.06.
Malling Juel	2		27.02.	17.03.	02.04.	29.06.		10.03.	03.04.	12.04.	25.06.
	3		20.02.	15.03.	31.03.	26.06.		08.03.	02.04.	10.04.	20.06.
Ojebyn	2		01.03.	17.03.	13.04.	29.06.		08.03.	05.04.	15.04.	25.06.
	3		24.02.	15.03.	12.04.	26.06.		06.03.	03.04.	11.04.	20.06.
Tsema	2		02.03.	19.03.	05.04.	27.06.		10.03.	04.04.	13.04.	24.06.
	3		24.02.	13.03.	01.04.	22.06.		06.03.	01.04.	09.04.	18.06.
Triton	2		01.03.	17.03.	09.04.	23.06.		05.03.	05.04.	16.04.	20.06.
	3		26.02.	15.03.	07.04.	20.06.		03.03.	03.04.	10.04.	16.06.
Čačanska Crna	2		26.02.	16.03.	31.03.	24.06.		10.03.	05.04.	14.04.	23.06.
	3		22.02.	11.03.	29.03.	20.06.		08.03.	03.04.	10.04.	20.06.

**Table 2 plants-11-00866-t002:** (**a**) Physical attributes of clusters and berries of 13 studied black currant cultivars (2018). (**b**) Physical attributes of clusters and berries of 13 studied black currants cultivars (2019).

(a)													
Cultivar	Age of Shoots	Cluster Weight (g)	Cluster Length (cm)	Number of Flowers per Cluster	Number of Berries per Cluster	Fruits Set (%)	Berry Weight (g)	Berry Diameter (mm)	Length of Shoot (cm)	Number of Clusters Per Shoot	Yield per Shoot (kg)	Total Yield per Plant (kg)	% of Shoot Yield in Total Yield
Ben Sarek	2	10.1	5.0	6.8	6.1	88.8	1.7	14.1	69.0	21.2	0.21	2.34	27.4
	3	7.7	4.6	6.2	4.9	79.2	1.3	13.1	97.1	64.9	0.57		72.6
Ben Nevis	2	7.2	5.5	6.5	4.4	66.8	1.6	13.8	82.3	14.3	0.10	1.50	20.8
	3	5.9	5.1	6.2	4.0	63.2	1.2	12.6	120.2	67.1	0.40		79.2
Bona	2	13.4	4.4	7.1	6.4	90.6	2.1	15.7	72.8	15.6	0.21	2.11	29.7
	3	9.2	3.6	6.8	5.8	85.4	1.5	14.0	114.4	53.7	0.49		70.3
Ben Lomond	2	6.9	5.1	7.9	6.8	86.0	1.0	12.0	110.2	22.5	0.16	1.41	33.1
	3	5.8	5.1	7.5	6.3	84.4	0.9	11.3	148.2	54.7	0.32		66.9
Ometa	2	6.8	5.5	8.7	7.6	87.7	0.9	12.5	81.3	21.0	0.14	1.39	30.8
	3	5.8	5.2	8.3	7.7	92.0	0.7	11.9	132.6	54.9	0.32		69.2
Tenah	2	9.7	6.0	8.3	6.3	75.2	1.5	14.1	68.1	14.6	0.14	1.66	25.5
	3	7.9	5.6	8.2	5.8	71.4	1.2	11.9	120.6	52.5	0.41		74.5
Silmu	2	8.8	6.3	11.9	7.0	59.1	1.4	13.1	98.5	16.2	0.14	1.81	23.5
	3	8.2	5.5	10.0	6.2	61.7	1.2	12.3	153.4	56.5	0.46		76.5
Titania	2	11.0	6.3	9.8	9.0	91.5	1.2	13.1	82.4	16.7	0.18	2.28	24.3
	3	9.9	6.0	9.8	9.3	94.9	1.1	11.5	139.8	58.2	0.58		75.7
Malling Juel	2	10.2	7.2	8.8	7.8	88.3	1.3	12.8	95.7	28.4	0.29	2.55	34.0
	3	8.8	6.6	8.2	7.5	91.8	1.2	11.5	157.2	63.8	0.56		66.0
Ojebyn	2	7.8	5.3	7.0	6.5	91.9	1.2	12.3	74.3	9.6	0.08	0.75	30.1
	3	6.2	5.0	7.3	7.0	95.5	0.9	11.4	117.6	28.3	0.18		69.9
Tsema	2	12.4	6.8	12.5	9.0	72.2	1.4	13.2	106.1	25.3	0.31	2.34	40.0
	3	8.4	6.4	11.7	8.7	74.3	1.0	11.1	135.1	55.9	0.47		60.0
Triton	2	7.2	5.6	10.8	7.6	70.5	0.9	11.7	57.4	13.3	0.09	1.07	26.8
	3	6.1	5.6	10.0	6.5	65.0	0.8	10.5	126.6	42.6	0.26		73.2
Čačanska Crna	2	9.9	7.8	10.7	9.0	84.2	1.1	12.5	83.8	20.9	0.21	2.21	28.1
	3	8.7	7.6	10.3	8.3	80.6	1.0	11.2	129.2	61.3	0.53		71.9
LSD		1.8	1.1	3.6	2.6	10.3	0.3	1.2	28.4	12.8	0.09	0.38	
**(b)**													
Cultivar	**Age of Shoots**	**Cluster Weight (g)**	**Cluster Length** **(cm)**	**Number of Flowers per Cluster**	**Number of Berries per Cluster**	**Fruits Set** **(%)**	**Berry Weight (g)**	**Berry Diameter (mm)**	**Length of Shoot (cm)**	**Number of Clusters per Shoot**	**Yield per Shoot (kg)**	**Total Yield per Plant (kg)**	**% of Shoot Yield in Total Yield**
Ben Sarek	2	9.2	6.4	9.8	6.8	69.1	1.5	13.9	59.8	17.7	0.16	1.81	27.0
	3	7.2	6.3	7.5	6.1	80.9	1.2	12.1	92.5	61.0	0.44		73.0
Ben Nevis	2	7.5	5.2	8.0	5.1	63.9	1.5	13.6	74.7	19.7	0.15	1.68	26.4
	3	4.9	4.7	6.8	4.9	72.5	1.2	12.3	109.3	65.3	0.41		73.6
Bona	2	11.3	6.6	10.4	6.5	62.8	2.0	15.2	71.5	17.0	0.19	1.91	30.2
	3	8.4	5.3	9.3	6.0	65.2	1.4	12.6	106.4	53.0	0.45		69.8
Ben Lomond	2	7.6	8.9	8.5	7.5	88.0	1.1	12.1	104.1	23.3	0.18	1.45	36.4
	3	5.8	8.1	7.4	6.4	85.8	0.9	10.4	144.3	52.7	0.31		63.6
Ometa	2	7.4	7.2	10.2	6.7	66.2	1.2	12.6	77.7	11.0	0.08	1.16	21.2
	3	5.8	6.2	8.3	6.2	75.1	0.9	10.2	119.3	52.7	0.30		78.8
Tenah	2	8.9	6.9	9.6	6.5	67.6	1.5	13.4	65.2	27.7	0.25	1.73	42.8
	3	7.7	6.3	8.2	6.3	76.2	1.2	12.6	108.3	43.0	0.33		57.2
Silmu	2	7.0	5.9	10.9	7.6	69.7	0.9	12.0	93.9	16.3	0.11	1.10	31.1
	3	5.0	5.6	8.6	5.9	68.6	0.8	10.1	139.0	50.3	0.25		68.9
Titania	2	9.6	6.2	10.6	8.6	81.1	1.1	12.4	79.6	15.7	0.15	1.40	32.3
	3	5.7	5.2	7.9	6.2	78.5	0.9	10.9	132.1	56.0	0.32		67.7
Malling Juel	2	8.0	8.2	11.2	7.8	69.9	1.1	12.4	101.5	36.0	0.29	2.02	42.7
	3	5.8	6.7	9.1	5.8	64.1	1.0	11.2	149.1	67.0	0.39		57.3
Ojebyn	2	6.2	5.7	6.8	5.4	79.4	1.3	13.1	63.9	7.7	0.05	0.56	25.7
	3	5.6	5.3	5.9	4.9	82.0	1.1	11.8	103.2	24.7	0.14		74.3
Tsema	2	11.2	10.2	15.2	9.9	65.0	1.2	12.4	97.8	34.7	0.39	2.58	45.2
	3	8.8	9.0	10.6	7.9	74.6	1.1	11.7	125.4	53.3	0.47		54.8
Triton	2	7.4	6.8	8.4	6.1	72.7	1.4	13.4	59.3	21.0	0.15	1.40	33.1
	3	6.0	5.9	6.9	5.6	82.2	1.0	11.3	112.5	52.3	0.31		66.9
Čačanska Crna	2	8.2	9.4	10.8	8.2	76.2	1.1	12.6	77.5	19.3	0.16	1.47	32.6
	3	6.0	7.2	8.9	6.5	72.4	0.9	11.4	124.2	55.3	0.33		67.4
LSD		1.8	2.3	3.5	1.4	8.3	0.4	1.7	24.4	12.3	0.08	0.43	

Data are means of 5 replications; The differences between the cultivars were tested with Fisher’s least significant difference test (LSD) at *p* < 0.05, level.

**Table 3 plants-11-00866-t003:** Contents of primary metabolites of berries of 13 studied black currant cultivars in 2018 and 2019 year.

Cultivar	Age of Shoots	2018	Total Soluble Solids (%)	Total Acids (%)	Total Sugars (%)	Sweetness	Ascorbic Acid (mg)	2019	Total Soluble Solids (%)	Total Acids (%)	Total Sugars (%)	Sweetness	Ascorbic Acid (mg)
Ben Sarek	2′ leaf		11.6	2.1	8.4	4.0	139.1		13.2	2.4	9.2	3.8	110.9
	3′ leaf		10.6	1.9	7.2	3.8	147.3		11.6	2.5	8.9	3.5	121.3
Ben Nevis	2‘ leaf		11.2	2.0	7.6	3.8	158.3		14.1	2.6	10.5	4.0	140.8
	3′ leaf		9.9	1.8	6.9	3.8	114.2		12.3	2.8	7.3	2.6	112.2
Bona	2′ leaf		10.5	1.1	8.5	7.9	135.5		11.7	2.4	9.0	3.7	125.4
	3′ leaf		8.9	1.1	7.2	6.5	121.3		10.6	2.5	6.9	2.7	106.3
Ben Lomond	2′ leaf		13.4	1.6	8.2	5.1	144.5		13.8	1.6	8.5	5.3	139.5
	3′ leaf		11.3	1.4	7.9	5.6	124.3		11.9	1.9	7.5	3.9	149.6
Ometa	2′ leaf		16.1	1.8	13.2	7.5	126.7		18.4	3.0	13.6	4.6	149.1
	3′ leaf		13.6	1.6	10.2	6.4	124.3		15.3	3.2	10.3	3.2	123.6
Tenah	2′ leaf		13.5	1.8	9.7	5.3	137.3		13.6	1.9	9.1	4.7	128.5
	3′ leaf		11.6	1.7	8.6	5.1	142.3		11.1	2.1	8.1	3.9	111.3
Silmu	2′ leaf		13.5	1.4	9.1	6.4	132.7		16.2	2.0	11.5	5.8	137.3
	3′ leaf		10.6	1.2	8.6	7.2	124.6		13.6	2.1	10.2	4.9	113.5
Titania	2′ leaf		12.9	1.3	9.1	7.0	121.3		13.7	1.4	9.3	6.9	145.4
	3′ leaf		9.6	1.2	8.2	6.6	111.5		11.2	1.7	7.9	4.8	136.3
Malling Juel	2′ leaf		13.9	1.6	10.0	6.3	153.7		14.8	2.0	10.4	5.1	144.3
	3′ leaf		10.6	1.8	9.2	5.1	132.3		12.3	2.3	8.2	3.5	174.5
Ojebyn	2′ leaf		12.7	1.5	7.1	4.8	122.4		14.1	2.8	10.9	4.0	109.1
	3′ leaf		9.9	1.4	7.2	5.1	112.3		11.6	2.8	7.6	2.7	89.3
Tsema	2′ leaf		13.7	1.5	9.0	6.0	169.0		13.4	2.1	9.9	4.7	147.8
	3′ leaf		11.1	1.4	8.1	5.8	123.5		10.9	2.3	7.2	3.1	124.3
Triton	2′ leaf		14.3	1.5	10.9	7.2	154.9		14.1	2.1	10.4	5.0	126.9
	3′ leaf		13.1	1.4	10.3	7.4	124.6		12.3	2.6	9.2	3.5	102.6
Čačanska Crna	2′ leaf		13.2	1.1	11.0	9.7	175.3		15.3	2.5	12.7	5.1	130.2
	3′ leaf		11.3	1.4	8.2	6.1	158.3		12.2	2.6	8.1	3.1	148.3
LSD			2.7	0.4	2.1		14.2		2.1	0.6	1.8		12.2

Data are means of 5 replications; The differences between the cultivars were tested with Fisher’s least significant difference test (LSD) at *p* < 0.05, level.

**Table 4 plants-11-00866-t004:** (**a**) Content of total phenols (mg/100 g f.w.), anthocyanins (mg/kg f.w.) and flavanols (mg/kg f.w.), aglycones, and DPPH (mg/mL) in 2018 year. (**b**) Content of total phenols (mg/100 g f.w.), anthocyanins (mg/kg f.w.) and flavanols (mg/kg f.w.) aglycones, and DPPH (mg/mL) in 2019 year.

(a)												
Cultivar	Age of Shoots	Total Phenolic Content	Total Anthocyanins Aglycones	Delphinidin 3-Rutinoside	Delphinidin 3-Glucoside	Cyanidin 3-Rutinoside	Cyanidin 3-Glucoside	DPPH	Kempferol	Myricetin	Quercetin	Total Flavonols
Ben Sarek	2	196.3	465.8	198.2	88.2	165.3	14.1	4.2	1.9	3.1	7.6	12.6
	3	154.3	348.5	145.3	45.8	148.2	9.2	5.2	1.7	3.3	6.3	11.2
Ben Nevis	2	154.2	649.1	226.3	112.3	245.3	65.2	4.2	2.1	3.2	7.3	12.6
	3	126.3	491.5	206.2	98.1	145.2	42.0	5.2	1.7	3.0	5.3	10.0
Bona	2	142.3	429.3	187.0	77.0	154.1	11.2	5.2	1.1	2.4	6.4	9.8
	3	122.3	325.5	134.1	42.1	137.0	12.3	5.7	1.0	2.1	5.4	8.5
Ben Lomond	2	198.6	658.3	228.6	114.6	247.6	67.5	2.6	1.2	3.1	4.5	8.8
	3	156.3	374.7	146.4	54.4	149.3	24.6	4.0	1.2	3.3	4.2	8.7
Ometa	2	202.3	765.8	289.2	127.2	301.2	48.2	4.1	1.0	2.1	8.1	11.2
	3	184.2	560.7	223.2	95.2	198.2	44.1	4.4	0.7	1.9	7.2	9.8
Tenah	2	123.0	425.0	175.0	40.0	187.0	23.0	5.2	1.2	2.8	7.9	11.8
	3	121.3	293.9	139.0	21.0	114.0	19.9	5.6	0.9	2.6	7.0	10.4
Silmu	2	136.2	467.7	196.6	86.6	163.7	20.8	4.5	1.2	2.7	5.3	9.2
	3	123.3	373.9	143.7	51.7	156.6	21.9	5.2	1.0	2.8	5.7	9.5
Titania	2	123.0	419.7	182.1	72.1	149.2	16.3	6.1	1.0	2.1	4.2	7.3
	3	111.2	382.8	185.3	37.2	142.1	18.2	7.2	0.6	2.5	3.7	6.8
Malling Juel	2	156.3	619.9	209.4	133.8	228.4	48.3	4.3	1.5	1.7	7.4	10.6
	3	142.3	404.7	127.2	103.6	130.1	43.8	4.8	1.3	2.1	6.9	10.3
Ojebyn	2	136.3	398.5	176.8	66.8	143.9	11.0	6.0	1.1	1.5	6.2	8.7
	3	123.3	361.6	180.0	31.9	136.8	12.9	6.3	0.9	1.9	5.3	8.0
Tsema	2	154.2	595.3	221.7	121.5	216.1	36.0	4.0	1.2	3.7	9.2	14.1
	3	123.5	355.5	114.9	91.3	117.8	31.5	5.6	1.1	3.0	8.2	12.3
Triton	2	141.3	426.9	162.6	81.0	158.1	25.2	5.2	1.1	1.7	5.1	7.9
	3	117.2	428.4	194.2	46.1	161.0	27.1	5.4	1.0	1.6	5.0	7.5
Čačanska Crna	2	189.5	524.1	243.0	124.0	218.6	38.5	4.1	0.9	2.6	6.6	10.0
	3	174.2	384.3	136.2	93.8	120.3	34.0	4.4	0.8	2.1	6.2	9.1
LSD		24.2	124.5	44.6	24.7	33.2	16.5	1.2	0.4	1.2	2.3	3.1
(**b**)												
**Cultivar**	**Age of Shoots**	Total Phenolic Content	**Total** **Anthocyanins Aglycones**	**Delphinidin 3-Rutinoside**	**Delphinidin 3-Glucoside**	**Cyanidin 3-Rutinoside**	**Cyanidin 3-Glucoside**	**DPPH**	**Kempferol**	**Myricetin**	**Quercetin**	**Total Flavonols**
Ben Sarek	2	228.3	753.6	311.2	142.3	278.6	21.5	2.8	2.1	3.0	6.7	11.8
	3	201.3	687.3	255.3	125.2	289.6	17.2	3.2	2.0	3.1	6.2	11.3
Ben Nevis	2	225.0	676.0	275.3	112.3	265.2	23.2	3.0	2.3	3.2	7.3	12.8
	3	212.3	441.9	201.0	98.2	123.5	19.2	3.1	2.1	3.0	6.4	11.5
Bona	2	202.3	708.8	300.0	131.1	267.4	10.3	3.0	1.3	3.2	8.2	12.7
	3	186.2	646.1	244.1	114.0	278.4	9.6	3.1	1.2	2.9	7.6	11.7
Ben Lomond	2	245.3	700.8	291.7	159.2	221.4	28.5	2.1	1.5	2.3	6.3	10.1
	3	233.6	575.6	250.4	127.9	180.1	17.2	2.3	1.2	2.1	6.1	9.5
Ometa	2	298.3	1160.8	475.3	265.2	333.1	87.2	2.4	2.1	2.1	8.5	12.7
	3	275.3	803.3	321.2	112.3	285.6	84.2	2.7	1.3	1.9	7.2	10.4
Tenah	2	198.2	454.0	261.1	51.0	118.9	23.0	3.6	2.3	2.9	8.3	13.5
	3	175.6	363.8	177.0	31.2	141.4	14.2	4.0	1.5	2.7	7.0	11.2
Silmu	2	178.3	747.2	309.6	140.7	277.0	19.9	3.1	1.1	3.1	6.3	10.5
	3	154.2	684.5	253.7	123.6	288.0	19.2	3.3	1.0	2.3	5.3	8.6
Titania	2	179.5	659.2	255.1	126.2	262.5	15.4	4.0	1.2	3.1	4.1	8.4
	3	165.3	603.0	209.2	109.1	273.5	11.2	4.0	1.0	3.2	4.0	8.2
Malling Juel	2	254.3	634.0	272.5	140.0	202.2	19.3	3.1	1.4	2.7	7.3	11.4
	3	222.2	515.8	231.2	108.7	160.9	15.0	3.9	1.3	2.8	7.2	11.3
Ojebyn	2	196.2	648.6	260.4	120.9	257.2	10.1	4.1	1.0	2.5	6.6	10.0
	3	166.3	583.8	203.9	103.8	268.2	7.9	4.3	0.8	2.6	5.3	8.7
Tsema	2	224.0	609.4	260.2	127.7	189.9	31.6	2.7	1.2	3.2	7.3	11.7
	3	201.2	505.8	188.9	96.4	193.2	27.3	3.6	1.4	3.1	7.1	11.6
Triton	2	201.3	677.0	274.6	135.1	243.0	24.3	3.0	1.0	2.3	5.3	8.6
	3	186.9	622.2	199.7	118.0	282.4	22.1	3.1	0.9	2.1	5.7	8.6
Čačanska Crna	2	234.3	585.9	310.2	33.7	225.3	16.7	3.0	1.0	2.3	7.8	11.0
	3	213.5	482.8	282.4	53.5	121.4	25.5	3.3	0.9	2.5	8.2	11.6
LSD		36.4	214.4	39.8	18.9	44.1	14.4	0.6	0.7	0.9	2.3	3.9

Data are means of 5 replications; The differences between the cultivars were tested with Fisher’s least significant difference test (LSD) at *p* < 0.05, level.

## Data Availability

All data obtained is contained in this article.

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
