# Peer review of "Does Shoot Age Influence Biological and Chemical Properties in Black Currant (Ribes nigrum L.) Cultivars?"

_plants, 2022, doi:10.3390/plants11070866_

Round 1
Reviewer 1 Report
Dear authors,
please use the Microsoft Word template or LaTeX template to prepare your manuscript.
Kind regards,
Boris
Author Response
Dear Reviewer,
an authors followed your instructions and used the Microsoft Word template to prepare manuscript.
Best Regards

Reviewer 2 Report
This manuscript summarizes results of a many-sided research that studies connections between fruit ripening, metabolite accumulation and antioxidant activity of 12 Ribes nigrum cultivars (represented by shoots of different ages). These results may be of interest to the readers of the “Plants”.
However, minor improvements are required before publication (according to the points 1-5):
1) Secondary metabolite composition, i.e., total phenol content, anthocyanins and flavonols were analyzed in the berries. Results of this phytochemical study are significant; however, these results were not compared with previous literary data. Authors should add a short caption to the “Discussion” section, comparing their quantitative data with those known in the literature. After this comparison, the novelty of the phytochemical study can also be highlighted.
2) Page 2, line 50: “…Yearly production is ~180,000 tones, and it is estimated….” Does it mean fruit amounts?
3) Page 2, lines 55, 56: “…Besides berries, seeds and leaves are rich source of numerous bioactive ingredients, especially polyphenolic compounds…” Reference is missing.
4) Page 2, lines 63-65: “…The development of shoots is influenced by environmental factors, such as soil temperature and moisture content, as well as by genetic factors such as heredity, and interactions between different parts of the plant…” Meaning of this sentence is not clear: how to interpret that “interactions between different parts of the plant” can influence shoot development? What type of interactions does this mean?
5) Page 2, lines 77, 78: “…The nutritional, organoleptic, and pharmaceutical qualities of its fruits represent a great source of natural antioxidants, vitamins and phenols…” This sentence is difficult to understand it should be improved.
6) Tables 2a, 2b: “Yield per shoot” is given in kg. Is this unit (kg) OK?
7) Page 8, Title of the Section is: “…Determination of total soluble solids (TSS), total aciditivity (TA), total sugars (TS) and ascorbic acid (AA)….” “Aciditivity” should be corrected.
In conclusion the manuscript deserves publication after a minor revision, detailed above.
Reviewer 3 Report
Dear Authors,
The manuscript entitled “Does shoot age influence biological and chemical properties in black currant (Ribes nigrum L.) cultivars?” could be published in Plants.
The aim of the study was to examine the influence of shoot age on biological and chemical properties of 13 black currant cultivars with different origin and ripening time. Phenological observations together with examined pomological and chemical characteristics were studied in two consecutive years at the experimental field near Belgrade, Serbia.
The manuscript is interesting, well written and designed. The manuscript seems suitable for publication according to the reviewer, only few points may require slight clarification.
Different methods used in the investigation could be included in the abstract.
The authors should choose one abbreviation for ascorbic acid (AA) or vitamin C (vit C) (see Table 3).
The authors could provide HPLC-UV chromatograms at different wavelengths, as well as MS/MS spectra of the main identified compounds. This will improved the manuscript.
